

# Optimized design of flux chambers for measurement of ammonia emission after field application of slurry with full-scale farm machinery

Johanna Pedersen*[1], Sasha D. Hafner[1], Andreas Pacholski[2], Valthor I. Karlsson[3], Li Rong[3], Rodrigo
Labouriau[4], Jesper N. Kamp[1]

[1]Aarhus University, Department of Biological and Chemical Engineering, Gustav Wieds vej 10D, 8000 Aarhus C, Denmark
[2]Thünen-Institute for Climate-Smart Agriculture, Bundesallee 50, Braunschweig, Germany
[3]Aarhus University, Department of Civil and Architectural Engineering, Inge Lehmanns Gade 10, 8000 Aarhus C, Denmark
[4]Aarhus University, Department of Mathematics, Ny Munkegade 118, 8000 Aarhus C, Denmark

*Correspondence to*: Johanna Pedersen (jp@bce.au.dk)

**Abstract.** Field applied liquid animal manure (slurry) is a significant source of ammonia ($NH_3$) emission, which is harmful to the environment and human health. To evaluate mitigation options, reliable measurements of effects are needed. A new system of dynamic flux chambers (DFC) with high time resolution online measurements was developed. The system was investigated in silico with computational fluid dynamics and tested in three field trials, each trial assessing the variability after application with trailing hose at different scales: manual (handheld), 3-m experimental slurry boom, and 30-m farm-scale commercial slurry boom. For the experiments with machine application, parallel $NH_3$ emission measurements were made using an inverse dispersion modelling method (backward Lagrangian stochastic modelling, bLS). The lowest coefficient of variation of replicate DFC measurements was obtained with manual application, followed by the 3-m slurry boom, and lastly the 30-m slurry boom. Conditions in DFCs resulted in a consistently higher $NH_3$ flux than what was measured with the inverse dispersion technique but both methods showed a similar emission reduction by injection compared to trailing hose: 89% by DFC and 97% by bLS. The new measurement system facilitates $NH_3$ emission measurement with replication after both manual and farm-scale slurry application with relatively high precision, with a coefficient of variation of 5% among replicates with manual slurry application and 20% for farm-scale slurry application.

## 1 Introduction

Liquid animal manure (slurry) can be utilized as a valuable nutrient source for crop production, but it is also a significant source of ammonia ($NH_3$) and greenhouse gas emissions through the whole manure management chain (Uwizeye et al., 2020). Emissions of $NH_3$ negatively affect the environment and human health and reduce the fertilizer value of slurry. If not properly managed, there is a high potential for emission of $NH_3$ from field application of slurry.



Emission depends on several factors, including application technique, weather, and slurry and soil properties (Hafner et al., 2018; Huijsmans et al., 2016; Webb et al., 2010). But there are significant knowledge gaps on effects of factors that influence emission, including interactions. There has been an increased research effort to measure, quantify, and model emission (Beltran

et al., 2021; Hafner et al., 2019; Hassouna et al., 2022; Webb et al., 2021). Recent research has investigated emission factors to improve the accuracy and precision of the national emission reporting as well as mitigation possibilities to reduce emission. Reliable and quantitative emission measurements of effects are needed.

Different methods used to measure $NH_3$ emission after field application of slurry can roughly be sorted into two categories: micrometeorological and enclosure methods (Shah et al., 2006). Inverse dispersion methods are micrometeorological methods

that can yield accurate flux measurements as they do not alter or manipulate the emitting area (Lemes et al., 2023) and are compatible with slurry application by farm-scale machinery (Kamp et al., 2021). With micrometeorological methods replication is usually omitted because of the scale of the plots and cross contamination between plots, making estimation of precision and statistical comparisons difficult. In contrast, enclosure methods, such as dynamic flux chambers (DFC), require only a small plot area, making replication simpler (Sommer & Misselbrook, 2016). In most DFC studies slurry has been applied

manually (handheld), which is not always representative of application with farm-scale machinery, especially when there is an interaction between the slurry applicator and the soil (Hafner et al., 2023.; Kamp et al., 2023). Therefore, to evaluate different application methods with repetitions, a system of chambers that can be used with farm-scale machine application of slurry is needed.

The goal of the present study was to develop a new DFC system that can be used after slurry application by farm-scale

machinery as well as manual application. An earlier generation of wind tunnels with high-time resolution $NH_3$ measurements (described in detail in Pedersen et al. (2020)) was found to have a coefficient of variation of 13% within triplicates with manual slurry application. The earlier generation wind tunnels can only be used with manual application as the tunnels are mounted on a metal frame that needs to be inserted into the soil. The installation of the metal frame after application by machine will cause slurry to spread out in the areas where the frame and slurry comes in contact and possibly alter the slurry exposed area

and slurry infiltration, and hence the emissions. The new system should provide high-time resolution flux estimates and have a lower inherent variation than the earlier generation, as well as being able to measure emissions after application by machine. To evaluate the new system design before construction in silico computational fluid dynamics (CFD) simulations were used for assessment of turbulence intensity over the soil surface. The CFD was employed to simulate the flow patterns in the DFC under different design scenarios, to avoid relying solely on time-consuming trial-and-error procedures. CFD has gained

widespread popularity for designing and optimizing the geometry of various devices and components (Scotto di Perta et al., 2016; Silva et al., 2022). The aim was that the chamber design should ensure a high degree of mixing of the air within the chamber and homogeneous air flow over the soil surface without 'dead areas' (without flow) or areas with very intense turbulence compared to the rest of the plot-area.

After construction, the system was tested and evaluated in three field trials in order to i) compare to the earlier generation of

wind tunnels described by Pedersen et al. (2020), ii) compare flux and relative differences between two application methods





measured with DFC and inverse dispersion modelling using the backward Lagrangian stochastic (bLS) model, and iii) quantify differences in precision after application of slurry with three different methods (manual, 3-m experimental slurry boom, and 30-m commercial slurry boom).

## 2 Materials and methods

The new DFCs were designed with inspiration from the laboratory chambers used by Dominique et al. (2013) and Georgios et al. (2013). An initial design was investigated thoroughly in silico with CFD to assess the homogeneity of airflow above the emission surface and to evaluate the level of turbulence intensity within the chamber. A prototype was built after positive CFD assessment and the recovery of $NH_3$ was measured with several different sample-air inlet designs. The final assessment was conducted with three field trials on the performance of the chambers for measurement of $NH_3$ emission after application of

slurry manually, by farm-scale machinery with a 30-m slurry boom, and by machinery with a smaller 3-m experimental slurry boom. For the application with machinery, measurements with inverse dispersion modelling were performed in parallel for comparison.

### 2.1 Dynamic flux chamber

### 2.1.1 Chamber design

Inspired by the laboratory chambers used by Dominique et al. (2013) and Georgios et al. (2013) a conceptual design of cylindrical chambers with a deflector plate was chosen (Fig. 1 and S1). With application by farm-scale machinery the chamber inlet air can have elevated concentrations of $NH_3$ when taken close to the soil surface. Therefore, inlets are positioned 1.4 m above soil surface to ensure a $NH_3$ concentration difference between air entering the chambers (background air) and the chambers' outlet air.

The emission chambers are made of slightly conical open-bottom polyethylene (PE) cylinders, with a thickness of 5-6 mm, a diameter at the bottom of 700 mm, and a height of 392 mm, giving a field plot area of 0.38 $m^2$. A deflector plate (6 mm plywood) was inserted 92 mm from the top with 30 mm between the edge of the plate and the sides of the chambers. Three galvanized steel pipes (1000 mm length, 80 mm diameter) are evenly distributed at the top of the chamber as air inlets. In the middle of the chamber air is drawn in with a fan (VH 125 hætte, Lindab A/S, Viby, Denmark). An iris diaphragm (DIRU 125,

Lindab A/S, Viby, Denmark) is located between the chamber and the fan to control and measure the AER, which is maintained at a fixed value during each trial.

For $NH_3$ analysis, sample air from the chamber is drawn at 1.5 L min$^{-1}$ through 15 m polyvinylidene difluoride (PVDF) tubing (OD: 6.35 mm, ID: 4.76 mm, Adtech Polymer Engineering Ltd, Stroud, United Kingdom). PVDF has been shown to have only minor $NH_3$ adsorption (Vaittinen et al., 2014). The tubing is insulated and heated to a minimum of 50°C. The tubing

length was 15 m so that the chambers can be used for experiments with farm-scale application machinery, which typically





spreads slurry with 24-30 m slurry-booms in Denmark. For background concentrations, three tubes are attached to an air inlet at three different chambers to measure the concentration of $NH_3$ in the air entering the chambers.

The sampling point for outlet air was designed to optimize mixing of the air before entering the sample tube (Fig. S3, Section 2.1.3). Between the PVDF tube and air sampling point a PTFE filter (47 mm diameter, 0.2µm pore size, Bohlender BmbH,

Grünsfeld, Germany) is inserted to ensure that no dust or other particles enter the tubing, valve, and analyzer.

To avoid air entering the chamber from the soil surface outside the area covered by the chamber, a plastic ring (OD: 750 mm, 30 mm heigh) is placed outside the chamber after the chamber has been placed in the field, and the space between the chamber and the ring is filled with sand (Fig. S4).

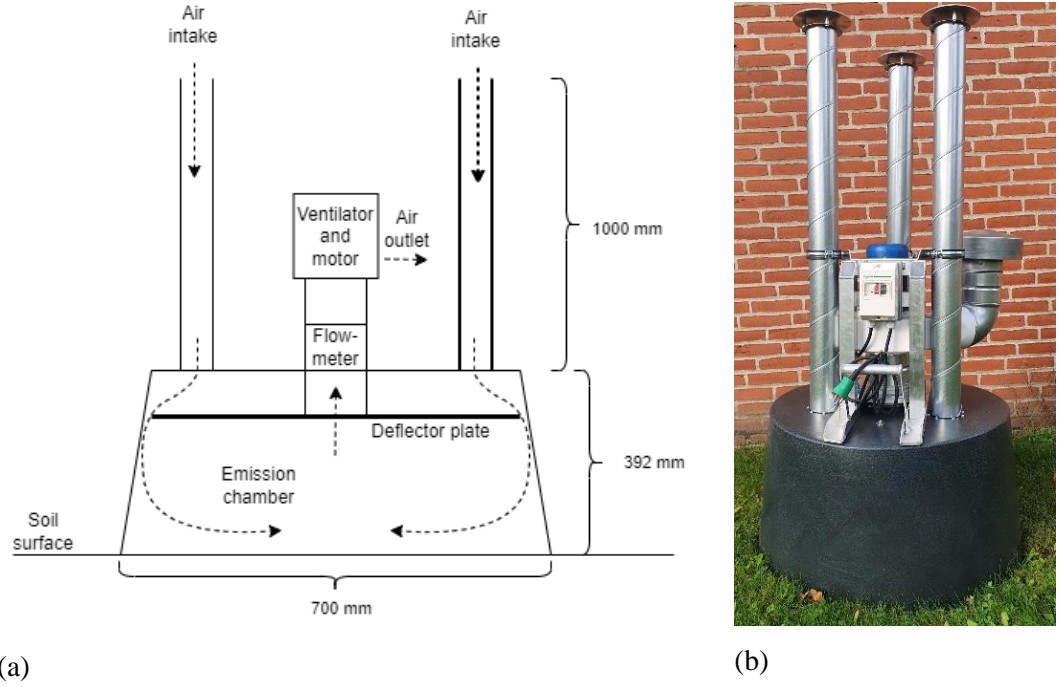

(a) (b)

**Figure 1: (a) Sketch of dynamic flux chamber, not to scale. (b) Picture of dynamic flux chamber. See also Fig. S1.**

### 2.1.2 Computational Fluid Dynamics

CFD modelling, conducted using the commercial software STAR CCM+ (STAR CCM+, 2020), was used to evaluate the uniformity of airflow and the level of turbulence intensity with four different configurations of the chambers (height of

deflector plate × AER). The geometric domain employed for the CFD modelling is illustrated in Fig. 2(a), and the mesh distribution in the vertical central plane (Z = 0.0 m) is displayed in Fig. 2(c), where XZ plane is located in the central plan of the chamber and 0 of Y starts from the floor of the chamber. To generate the mesh within the computational domain, a





combination of polyhedral and prism meshers was used to generate the mesh. The prism layers were generated near solid

surfaces to resolve the boundary layer properly. The first layer of the mesh was placed in the position where y+ value was

either larger than 30 or smaller than 5 to fulfill the requirement of the two-layer y+ treatment wall function. The final number

of the mesh utilized in the simulations was 342,354 for simulations A and B and 987,519 for simulations C and D.

The RANS (Reynolds-averaged Navier-Stokes) method was used. The realizable two-layer k-e model was adopted to simulate

the turbulent kinetic energy and dissipation rate of turbulent kinetic energy. The two-layer y+ wall treatment was selected. The

second accuracy upwind scheme was applied to discretize the convection terms in partial differential equations (PDEs). The

convergence criteria were set as $10^{-3}$ for continuity, three velocity components and two turbulent quantities. Additionally, the

air velocities at several points were also monitored to assist assessing the convergence of the simulations.

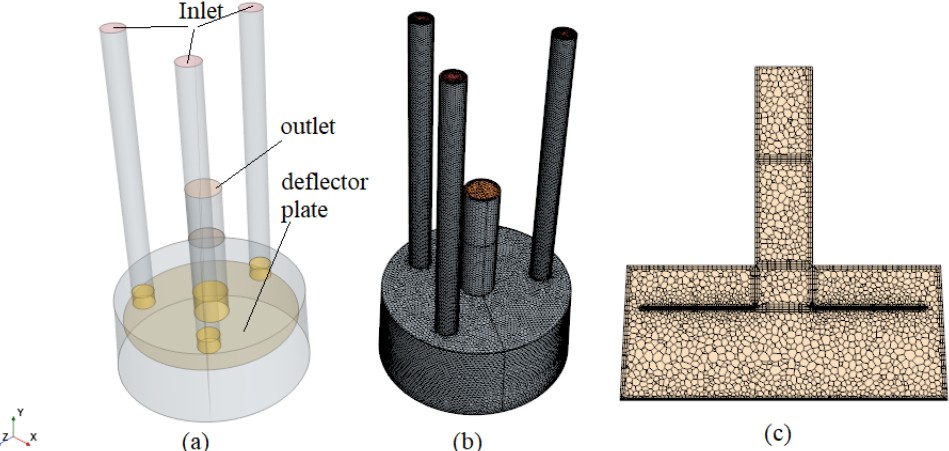

**Figure 2. Schematic configuration and illustration of mesh distribution of case A. (a) geometry model, (b) surface mesh and (c) mesh**
**distribution at plane of Z = 0.0 m.**

The CFD simulations were conducted in an isothermal condition. A no-slip condition was imposed at all solid surfaces. The

inlet was set as a velocity inlet with turbulent intensity of 10% and turbulent length scale of 0.0056 m. The outlet was defined

as a pressure outlet with pressure of 0.0 Pa. The boundary conditions for CFD simulation scenarios can be found in Table 1.


**Table 1. Boundary conditions for CFD simulation scenarios. AER: Air exchange rate.**

| Simulation | Height of deflector plate[1] (m) | Volume (m³) | AER (min⁻¹) | Inlet air speed (m s⁻¹) |
|---|---|---|---|---|
| A | 0.2 | 0.111549 | 10 | 1.234 |
| B | 0.2 | 0.111549 | 15 | 1.850 |
| C | 0.3 | 0.150014 | 15 | 2.488 |





| D | 0.3 | | 0.150014 | 20 | 3.318 |
|---|-----|--|----------|----|-------|

[1]Distance from soil to the plate

### 2.1.3 Recovery of ammonia, mixing within chamber, and stability of airflow

All concentration measurements for the evaluation of the chambers were done with a cavity ring-down spectrometer (CRDS) (G2103 $NH_3$ concentration Analyzer, Picarro, CA, USA).

To evaluate stability of measurement and recovery, $NH_3$ emission was measured from a standard solution of $NH_4Cl$ (4 g N L$^{-1}$) using several different sampling air inlet designs. Furthermore, the time needed for a stable concentration reading on the CRDS and the stability of the reading at two different AER (15 and 20 min$^{-1}$) was evaluated after manual application of cattle slurry in bands mimicking trailing hose application.

For recovery and stability evaluation, 50 mL of the $NH_4Cl$ solution was added to an open container (length: 150, width: 110 mm, height: 20 mm). To induce emission, pH was increased to >10 by adding 1 mL 32% NaOH. The container was then immediately placed under the tunnel, air flow through the chamber was started, and $NH_3$ concentration was measured with the CRDS. Each trial lasted between 15 and 60 min. At the end of each trial 0.5 mL 96% $H_2SO_4$ was added to the $NH_4Cl$-solution to decrease pH (< 4), which stopped emission. Samples of the $NH_4Cl$ solution were taken before and after emission measurement to determine the loss of N by difference, which was then compared to the loss calculated from the concentration measurements with the CRDS and air flow. Three inlet designs were tested: single point, y-shaped with 15 quadratically spaced sampling points (Fig. S2, corresponding to C3u configuration from Table 1 in Loubet et al. (1999)), and a new design with the goal of optimized mixing. The new inlet design consisted of 100 mm PTFE tube (OD: 6.35 mm, ID: 4.75 mm) inserted into a 15 mL plastic centrifuge tube which was itself inserted into a 50 mL plastic centrifuge tube. All three (PTFE tube and both centrifuge tubes) had 3 rows of 5 small holes (Fig. S3).

In a preliminary trial, six DFCs were used with manual cattle slurry application to test the stability of concentration measurements, to determine if 8 min is sufficient to reach a stable reading and determine if AER affects emission dynamics. The volumetric air exchanges rates (AER) in the emission chamber were adjusted to 15 and 20 min$^{-1}$ to match the CFD simulations (Section 2.1.2), with three DFCs for each AER. Cattle slurry (1.5 L) was applied manually to a grass field in two bands to mimic trailing hose application, corresponding to an application rate of 30 metric tonne ha$^{-1}$. The measurements were conducted for 60 h and the average temperature was 9.1°C. See supporting material for results.

### 2.2 Field trials

Three field trials were conducted in October and November 2022 at Campus Viborg, Aarhus University (Table 2). Anaerobically digested slurry (digestate) was applied by three different application systems (manual, 3-m experimental slurry boom, 30-m farm-scale commercial slurry boom) to assess the variability between replicates with different slurry application





strategies. The trials were conducted in three different periods in the same field. Emission was measured for 60 h for trial A and 120 h for trials B and C using the DFC. Emission data and calculations for the field trials can be found at: http://github.com/AU-BCE-EE/Pedersen-2023-DFC.

### 2.2.1 Overview of field trials

In trial A (Table 2), digestate was applied manually in bands at the soil surface with a hose connected to a waterings can with the predetermined volume of digestate. In parallel, the emissions were measured with a setup of an earlier design of wind tunnels (WT) described in Pedersen et al. (2020). 9 DFC and 3 WT were included in a block design, each block with 3 DFC and 1 WT (Fig. S5 and S6). The total of 12 chambers (DFC and WT) were all connected to the same valve and same instrument for measurements of $NH_3$ concentration, which eliminates the risk of biases between instruments.

For application in trial B (Table 2), a commercial 30-m trailing hose (hose diameter: 50 mm) boom (SB series, Samson Agro A/S, Viborg, Denmark) was used for digestate application. The driving speed during the application was approximately 7-8 km h$^{-1}$. The digestate was applied in a square $30 \times 70$ m plot. Immediately after application 8 DFC were placed in the plot (Fig. S7-S9) and measuring points for bLS were placed inside and upwind of the plot. In total 3 CRDS instruments were used in trial B.

In trial C (Table 2), digestate was applied in two plots by 3-m booms specially constructed for field trials. In one plot digestate was applied by trailing hose (hose diameter: 50 mm) and in the other by injection. The driving speed during application was approximately 3.5-4 km h$^{-1}$. Immediately after application 6 and 7 DFC were placed in the plots for trailing hose and injection application, respectively (Fig. S10-S13) and a measuring point for bLS was placed inside each of the two plots and with one upwind position. In total 4 CRDS instruments were used in trial C.

Table 2. Overview of field trials. DFC: dynamic flux chambers, WT: wind tunnels, bLS: backward Lagrangian Stochastic model.

| Trial | Application time | Application technique | Application method | Plot area | Measuring method | Number of chambers |
|-------|-----------------|----------------------|-------------------|-----------|-----------------|-------------------|
| A | 2022-10-11 14:57 | Trailing hose | Manual | | DFC, WT | 9 DFC, 3 WT |
| B | 2022-11-16 09:42 | Trailing hose | 30-m boom | 100 m$^2$ | DFC, bLS | 8 |
| C | 2022-11-24 11:18 | Trailing hose | 3-m boom | 56 m$^2$ | DFC, bLS | 6 |
| C | 2022-11-24 11:42 | Injection | 3-m boom | 47 m$^2$ | DFC, bLS | 7 |

### 2.2.2 Digestate and soil properties and climatic conditions during the trials

The digestate was produced at the biogas plant at Aarhus University, which operates two reactors in series at 51°C for 14 d and 47°C for 40 d. After the second reactor, the digestate was pumped to a concrete storage tank, where the digestate for the



trials was collected. The input to the first reactor in the period where the digestate was produced for the trials was 82% mixed cattle and pig slurry, 9% deep litter, and 9% grass and grass silage (by fresh mass).

The digestate was analyzed using standard methods for dry matter (DM) content (American Public Health Association, 1999), total nitrogen (Association of Official Analytical Chemists, 1999), and total ammoniacal nitrogen (TAN) (International Standard, 1984). All digestate properties and application rates during the trials can be found in Table 3.

**Table 3. Digestate properties (± standard deviation ($n = 2$)) and application details. TAN: total ammoniacal nitrogen, DM: dry matter, N: nitrogen, DFC: dynamic flux chambers, WT: wind tunnels.**

| Trial | Application rate (kg TAN ha$^{-1}$) | Application rate (tonnes ha$^{-1}$) | DM (%) | Total N (g kg$^{-1}$) | TAN (g kg$^{-1}$) | pH |
|---|---|---|---|---|---|---|
| A | 90 (DFC), 70 (WT) | 45 (DFC), 35 (WT) | 5.99 ± 0.02 | 2.62 ± 0.40 | 2.00 ± 0.02 | 7.5 ± 0.01 |
| B | 64 | 35 | 5.43 ± 0.09 | 2.61 ± 0.01 | 1.83 ± 0.09 | 7.5 ± 0.01 |
| C | 67 | 35 | 5.03 ± 0.03 | 2.73 ± 0.14 | 1.93 ± 0.07 | 7.6 ± 0.01 |

The field had a loamy sand texture with barley stubble. Gravimetric water content and dry bulk density were determined using 100 cm$^3$ soil cores taken at 0-5 cm depth. The dry bulk density was 1.30 ± 0.10 g cm$^{-3}$ ($n = 9$). At the beginning of each trial the gravimetric water content was 0.19 ± 0.002, 0.22 ± 0.01, and 0.22 ± 0.01 g g$^{-1}$ ($n = 3$) for trials A, B, and C, respectively.

Ambient air temperature was measured at a local weather station located < 0.5 km from the plot for trial A and logged in 1 h averages. For trials B and C, average temperature and wind speed in 30 min intervals were derived from 16 Hz measurements with an ultrasonic anemometer (WindMaster, Gill, Hampshire, UK) at 2 m height. Temperature and wind speed during the trials can be found in Fig. 4.

### 2.2.3 Emission measurements

For all chamber emission measurements (DFC and WT), NH$_3$ emission was measured continuously with a Cavity Ring-Down Spectrometer (CRDS) model G2103 (Gas Concentration Analyzer, Picarro, CA, USA) whereas two or three model G2509 analyzers (Gas Concentration Analyzer, Picarro, CA, USA) were used for bLS measurements. These instruments have been shown to be robust and reliable in agricultural environments (Kamp et al., 2019).

### 2.2.3.1 Dynamic flux chambers and wind tunnels

For the DFC and WT measurements air was drawn to a 10-port rotary valve or two 10-port rotary valves connected to become one 19-port rotary valve (model C46R, C45R-8140EUTA, VICI, Valco Instruments Co. Inc., Houston, TX, USA). Measurements were done on each stream for 8 min, yielding a data point every 144 min for trial A (9 DFC with 3 backgrounds and 3 WT with 3 backgrounds), every 88 min for trial B (8 DFC and 3 backgrounds), and every 80 min for trial C (7 DFC and 3 backgrounds per application method). During the trials the AER in the DFC was set to 20 min$^{-1}$.





Each WT consisted of an open-bottomed stainless-steel chamber (250 mm height, 800 mm length, 400 mm width). The airflow through the chamber was controlled with a fan, motor, and frequency converter which was connected to the emission chamber via a steel duct. The air inlet was a narrow 130 mm heigh slot. The AER was set to 25 min$^{-1}$. Each WT was mounted on a metal frame inserted into a soil, giving a field-plot area of 0.2 m$^2$. Sample air was drawn through heated PTFE tube (OD: 6.35 mm, ID: 4.75 mm). A detailed description of the WT can be found in Pedersen et al. (2020).

For both DFC and WT an average of the last 30 s of measurements per 8-min measurement cycle was used for calculations of the flux and cumulative emission. An average of the background measurements ($n$ = 3 locations) was subtracted for each measurement cycle concentration before further calculations. The background-corrected concentration ($C$, mg m$^{-3}$), the air flow in the emission chamber ($q$, m$^3$ s$^{-1}$), and the area of the soil surface covered by the tunnel ($A$, m$^2$) were used to calculate the flux ($F$, mg s$^{-1}$ m$^{-2}$) (Equation 1).

$$F = \frac{C \cdot q}{A} \tag{1}$$

The contribution from each measurement cycle to the cumulative emission was calculated from the flux using the trapezoidal rule.

**2.2.3.2 The backward Lagrangian Stochastic method**

An inverse dispersion model, the backward Lagrangian stochastic (bLS) model, was used to obtain NH$_3$ fluxes in trails B and C. The bLS model has previously been used to estimate NH$_3$ emissions after slurry application (Hafner et al., 2023.; Kamp et al., 2021) and it simulate air movement backwards in time from the origin of the sensor based on the wind field in a certain averaging interval, in these trials 30 minutes intervals. The bLS model was used with the R software package bLSmodelR (https://github.com/ChHaeni/bLSmodelR, version 4.3, (Häni et al., 2018)), which produces a concentration-to-emission ratio

(CE$_{bLS}$) for each averaging interval.

The flux (F) is calculated using the CE-value in combination with NH$_3$ concentrations measured up- ($C_{upwind}$) and downwind ($C_{downwind}$) of the source (Equation 2).

$$F = \frac{(C_{downwind} - C_{upwind})}{CE_{bLS}} \tag{2}$$

The inputs to the bLS model are the friction velocity (u*), the roughness length (z$_0$), the Obukhov length (L), the standard

deviation of the three wind components ($\sigma_u$, $\sigma_v$, $\sigma_w$), and the wind direction. In these two trials, 100,000 trajectories were used in the model calculations. The position of the source and sensor are also inputs to the model and the positions were mapped with a GPS (Trimbel R10, Sunnyvale, California, USA). A detailed explanation of the bLS model can be found in Häni et al. (2018).

The atmospheric conditions affect the accuracy of the bLS model, and the method is most accurate in conditions where the

245 Monin-Obukhov Similarity Theory reasonably can be applied (McBain & Desjardins, 2005). Thus, observations were filtered (removed) when u* ≤ 0.05 m s$^{-1}$, |L| ≤ 2 m, z$_0$ > 0.1 m, $\sigma_u$/u* > 4.5, and $\sigma_v$/u* > 4.5 (Bühler et al., 2021; Lemes et al., 2022). Flux was estimated by linear interpolation in these intervals for calculation of cumulative emission.





For trial C, the three CRDS instruments were used to measure ambient $NH_3$ in the field before digestate application. The concentration for all three instruments were stable for 9.5 h before application, and there was a small offset between the instruments. In that timespan, the instruments average background concentrations were $1.920 \pm 0.06$ µg $NH_3$ m$^{-3}$, $0.273 \pm 0.03$ µg $NH_3$ m$^{-3}$, and $0.824 \pm 0.11$ µg $NH_3$ m$^{-3}$ for the instruments later used for the injection plot, trailing hose plot, and background, respectively. The instrument for the injection plot and trailing hose plot were corrected for the offset by subtracting 1.096 µg $NH_3$ m$^{-3}$ and adding 0.551 µg $NH_3$ m$^{-3}$, respectively. The difference was very small compared to the measured background concentrations of $NH_3$, but for this trial, the concentration difference between background and the plots were low, thus the small offset would have influenced the fluxes from the injection plot if not corrected.

**2.2.4 Statistics for comparing emission measurements**

Emission measurements were compared in multiple ways. Emission measurements were normalized by applied TAN and expressed as a percentage lost per minute and compared graphically for each trial. Total cumulative emission was expressed as a percentage of applied TAN. For methods with replication (DFC and WT) a mean value, standard deviation, and coefficient of variation (cv, standard deviation divided by the mean) was calculated from the individual estimates determined with each separate chamber.

Differences in variability or precision between DFC and WT as well as among application methods were primarily assessed by comparing cv values. Also, a detailed analysis of the flux over time from DFC and WT with manual application was performed using a Gamma generalized linear mixed model defined with the logarithmic link function and a random component representing the tunnel or the chamber and a factor representing the elapsed time (following a statistical methodology similar to that in other emission studies (Pedersen et al., 2021a; 2021b). Separate models were adjusted for the WT and DFC measurements, facilitating a comparison of the variability obtained with the two measurement methods by comparing the estimated dispersion parameters. Since the variances of the responses under a Gamma model are proportional to the square of the expected value of the response, with a proportionality coefficient given by the dispersion parameter (which was different for the two methods), the variance of the two methods was compared by examining the two curves relating the predicted variance to the predicted means.

Trial C provided an opportunity to compare application method effects measured with the new DFC design to bLS measurements. Replicates DFC measurements from trailing hose application and injection were used together to calculate a 95% confidence interval on the relative emission reduction provided by injection. For this task the unit of analysis was an individual DFC and the response variable was total cumulative emission as a percentage of applied TAN. The t.test() function in R (stats package, v4.2.1, R Core Team 2023) was used with log$_{10}$-transformed values to simplify calculation of a relative effect. Estimated 95% confidence limits based on separate estimates of variance for each group and the Welch modification to degrees of freedom were back-transformed to express the reduction as a percentage of trailing hose emissions. A bootstrap approach based on resampling was also applied for comparison (Davison and Hinkley).



The bLS measurements were not replicated, making it difficult to estimate random error. An important source of error for the

bLS method is bias between instruments, especially when the concentration differences are small, as with the injection plot in

particular. Therefore, an estimate of uncertainty related to potential instrument bias in concentration measurements was made.

Although this estimate does not include all potential sources of error, because the two plots were in the same field and

measurements occurred at the same time, other sources of error are likely to be similar for the two plots, and therefore less

important for estimation of a relative difference. A standard deviation in overall concentration bias was based on background

measurements made by the three instruments over a 9.5 h period prior to digestate application (Section 2.2.3) combined with

measurements from a 30-h period 10 days after application, when any effect of slurry application was expected to be negligible.

Because there is no replication for concentration measurements, this standard deviation directly provides an estimate of how

bias may vary for a randomly selected instrument. Because emission calculations depend on the concentration difference

(Equation 2), a value for the difference between two instruments ($s_{bd}$, for standard deviation of the 2-instrument bias in

measurement of a concentration difference) was estimated using the formula for independent errors (Equation 3.13 in Taylor,

1982). Uncertainty in an individual emission measurement interval can then be calculated by using this value as the numerator

in Equation (2). Assuming constant bias over a trial, overall standard deviation in measured emission was estimated using

Equation 3, where $CE_{bLS}$ = an average concentration-to-emission ratio over the emission measurement trial duration, $\Delta t$ =

interval duration (0.5 h = 1800 s), and $n_{int}$ = number of intervals (240).

$$s_{emis} = \frac{s_{bd}}{CE_{bLS}} \Delta t \cdot n_{int} \qquad (3)$$

This estimate of standard deviation in measured emission was applied to both plots in a parametric bootstrap approach in order

to estimate of a 95% confidence interval for the relative reduction (Davison and Hinkley, 1997).

### 3 Results and discussion

### 3.1 Computational Fluid Dynamics velocity magnitude

Fig. 3 displays the velocity magnitude, ranging from 0.0 m s$^{-1}$ to 0.5 m s$^{-1}$, distribution at a horizontal plane 0.05 m above the

soil. When AER was 10 min$^{-1}$ the velocity magnitude was smaller than 0.17 m s$^{-1}$ (Fig. 3(a)). When AER was 15 min$^{-1}$ and the

deflector plate was 0.2 m above the soil surface, the velocity magnitude was up to about 0.15 m s$^{-1}$ if the relatively higher air

speed near the edge of the chamber was not considered. With the deflector height of 0.3 m above the soil surface, the average

velocity magnitude of the plane was about 0.2 m s$^{-1}$ (Fig. 3(c)). Although the deflector plate was placed 0.10 m higher in Fig.

3(c) compared to that in Fig. 3(b) with the same AER of 15 min$^{-1}$, the inlet air speed was 2.49 m s$^{-1}$ in case of Fig. 3(c) while

it was 1.85 m s$^{-1}$ in case of Fig. 3(b). Thus, the velocity magnitude was still slightly higher in Fig. 3(c). As the AER was

increased to 20 min$^{-1}$ in the case shown in Fig. 3(d) with deflector plate height of 0.3 m, the velocity magnitude of the plane

was notably increased and up to 0.45 m s$^{-1}$. In order to ensure that the velocity was high enough for the air within the chamber

to be well mixed and avoid any 'dead zones' over the soil surface, the plate height of 0.3 m and an AER of 20 min$^{-1}$ was chosen

for the field experiments.





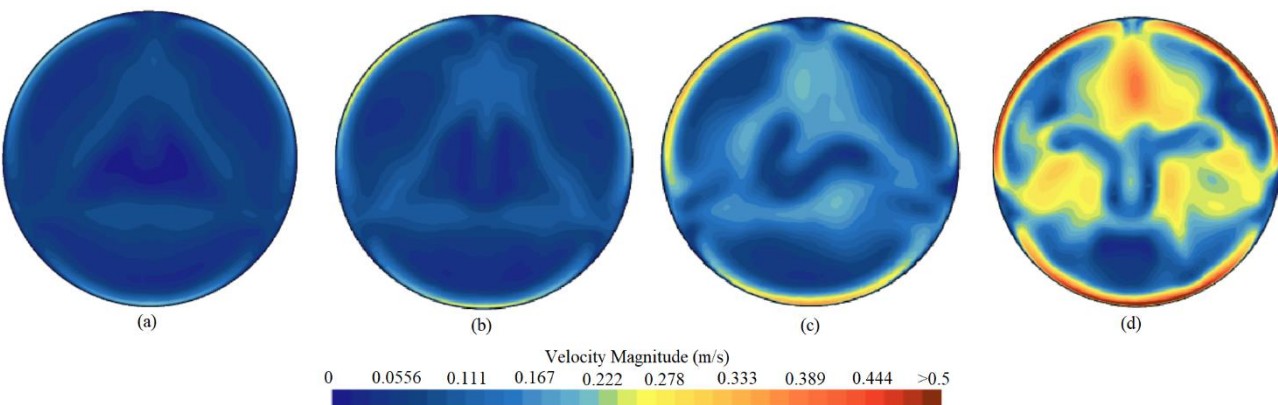

**Figure 3. Velocity distribution at a horizontal plane 0.05 m above the soil surface, (a) deflector plate height of 0.2 m and AER of 10 min⁻¹; (b) deflector plate height of 0.2 m and AER of 15 min⁻¹; (c) deflector plate height of 0.3 m and AER of 15 min⁻¹; and (d) deflector height of 0.3 m and AER of 20 min⁻¹.**

### 3.2 Recovery of ammonia, mixing within chamber, and stability of airflow

Both the single-point measurement and the y-shaped measuring inlet resulted in very unstable measurements where concentrations could change up to several hundred ppb (>35% of average) within a few seconds (data not shown). The final inlet design (Fig. S3) was found to give steady measurements in the range of concentrations found after slurry application (15-2500 ppb). The final inlet design gave an average recovery of emitted $NH_3$ of $102 \pm 1\%$ (mean ± standard deviation, $n = 3$) from the $NH_4Cl$ test. This high recovery shows that irreversible adsorption is insignificant in the concentration range measured during the recovery test (800-3000 ppb), however, at lower concentrations adsorption might cause longer response times.

The two different AERs did not yield different emission dynamics (Fig. S14), but emission was lower with the lower AER. Similar results have been observed earlier, where increasing AER increased emissions (Hafner et al., 2023). There was no difference in concentration measurement stabilization time with the two different AER (Fig. S15) and 8 min measurement was sufficient to reach a stable reading, except for the first round of measurements, which is a known issue due to adsorption of $NH_3$ on all surfaces in the measurement system (Pedersen et al., 2020).





## 3.3 Field trials

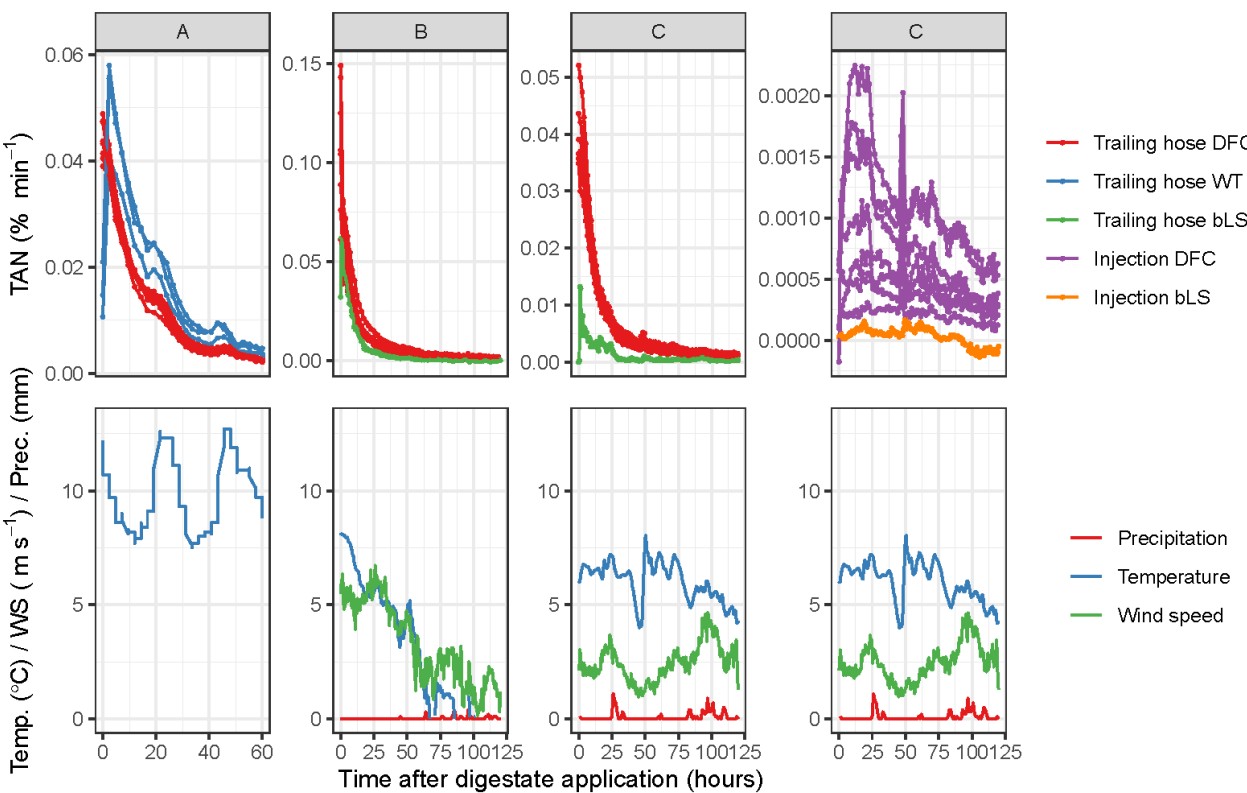

**Figure 4. Ammonia emission rate (as a fraction of applied TAN per minute) and weather after application of digestate by trailing hose or injection measured with dynamic flux chambers (DFC), wind tunnels (WT), or backward Lagrangian stochastic model (bLS). Digestate was applied manually (A), by 30-m boom (B), or 3-m boom (C).**

**Table 4. Cumulative ammonia loss (% TAN) measured with dynamic flux chambers (DFC), wind tunnels (WT), and backward Lagrangian stochastic model (bLS) after application of digestate by trailing hose or injection manually, with a 3-m boom, or with 30-m boom. Mean, number of observations for chamber methods (n), standard deviation (sd), and coefficient of variation (cv) is reported for DFC and WT.**

| | | | DFC results | | | | WT and bLS results | | | | |
|---|---|---|---|---|---|---|---|---|---|---|---|
| | Application | Application | Emission | | | | | Emission | | | |
| Trial | technique | method | (% TAN) | cv | sd | n | Method | (% TAN) | cv | sd | n |
| A | Trailing hose | Manual | 42.5 | 4.7 | 2.0 | 9 | WT | 60.7 | 14.0 | 8.5 | 3 |
| B | Trailing hose | 30 m boom | 62.6 | 19.5 | 12.2 | 8 | bLS | 42.7 | | | |
| C | Trailing hose | 3 m boom | 42.8 | 13.6 | 5.8 | 6 | bLS | 8.77 | | | |
| C | Injection | 3 m boom | 4.7 | 55.3 | 2.6 | 6 | bLS | 0.23 | | | |





### 3.3.1 Performance of dynamic flow chambers

The new DFC design performed better than the earlier generation WT design, with less of a delay or lag in initial emission
measurements and lower variability in emission among replicates. The DFC did not show the same lagging effect during the
first measurement as the WT (Fig. 4). Lagging in the first measurement of the WT will cause an underestimation in cumulative
$NH_3$ emission (Hafner et al., 2023.; Pedersen et al., 2020). The improvement with the new DFC design is likely due to better
heating of the sampling lines or the tube material (PVDF) that has a lower affinity for NH3 adsorption than PTFE (Vaittinen
et al., 2014). Furthermore, the improved sampling lines likely reduce the general lagging effect which causes slight under- or
overestimations of the $NH_3$ concentrations measured, on the order of 1-3.5% for tubes heated to a lower temperature, depending
on the concentration of the preceding measurement (Pedersen et al., 2020).

The Gamma generalized linear mixed models adjusted well (*p*-values for a goodness of fit equal to 0.62 and 0.89 for the DFC
and the WT respectively, see Fig. S16 and S17 for quantile-quantile plots). The dispersion parameters were estimated as $2.2 \cdot 10^{-5}$
and $3.3 \cdot 10^{-5}$ for the models describing the DFC and the WT determinations, respectively. This indicates that the WT fluxes
were characterized by a substantially larger variability compared to the DFC (Fig. 5).

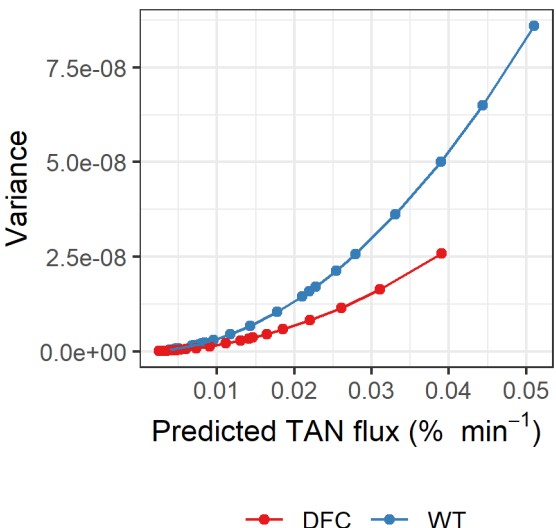

**Figure 5. Theoretical variance expressed as a function of the theoretical mean of $NH_3$ flux measurements with dynamic flux chambers (DFC) and wind tunnels (WT) after trailing hose application of digestate.**


Ammonia emission measurements after field application of slurry has at least three sources of variation: error inherent in
emission measurement systems, uneven slurry application, and heterogeneity in soil properties that affect emission. The last
two sources are expected to contribute to variation for measurements with enclosure methods to a higher degree than
measurements with micrometeorological methods because of their small plot size. As expected, the standard deviation and





coefficient of variation increased in the order: manual application < 3 m boom < 30 m boom (Table 4). This was expected as it is possible to evenly apply the slurry close to the soil surface when using manual application, resulting in a uniform application in all bands. When applying slurry by a 30-m boom it is expected that there can be differences in the amount of slurry leaving each hose, due to differences in length from the distributor, potential plugging, or other differences between hoses which will cause variation in the local application rate and exposed slurry surface area along the boom. Furthermore,

higher driving speed with the 30-m boom application compared to the 3-m boom application is expected to also cause higher variation in the exposed surface area of each digestate-band as the higher speed causes more movement of each individual trailing hose, especially over uneven soil surfaces. The same field was used, but the plot area between the trials differed in the present study (Fig. S5, S7, and S10). This indicates that the main cause of the variation is the application itself and not soil heterogeneity, because the soil heterogeneity is likely minor within the area where the tunnels are placed between the three

trials.

The differences in variation between the application methods indicates that it will be possible to detect much smaller differences between treatments if slurry is applied manually compared to machine application. Based on the standard deviation values (Table 4), for an equal sample size (number of DFCs), manual application could be used to detect a difference smaller by a factor of 5 to 6, compared to the 30-m boom. Although there is still some difference between manual and machine

application, manual application might be applicable when mimicking trailing hose application and slurry properties are the parameters investigated. The relatively large variation with machine application makes it difficult to compare different low-application techniques to each other. To properly compare any application methods where there is a direct physical interaction between soil and the application machinery (e.g., trailing shoes and injection) it is necessary to apply the slurry with farm-scale machinery to get a realistic response. In some cases, the differences between different low-emission application

techniques can be small (10-20% of emission; (Häni et al., 2016; Webb et al., 2010)), in which case it would most likely not be possible to detect statistically significant differences with a system such as the one presented in the present paper.

This high variation for application by machinery indicates that more measurement replicates (DFCs) are needed, compared to manual application. This difference should be carefully considered when planning field trials due to the trade-off between replicates and number of different treatments that can be tested.

Although this new DFC design was used only for $NH_3$ from field-applied slurry, it should also work for other sources and other compounds. Other DFC designs have been used for $NH_3$ from mineral fertilizer and VOC from slurry (Pedersen et al., 2021c; Scotto di Perta et al., 2020). Measuring less-adsorbing compounds than $NH_3$ could even allow for higher time resolution of the measurements.

**3.3.2 Comparison with inverse dispersion measurements**

The $NH_3$ flux and consequently cumulative emission measured with the DFC was much higher than bLS measurements in the two trials that included both measuring methods (trials B and C, Fig. 4, and Table 4). Other studies also found over- or underestimation of $NH_3$ emissions measured with flux chambers compared to micrometeorological methods (Hafner et al.,



2023.; Kamp et al., 2023; Mannheim et al., 1995; Misselbrook et al., 2005; Ryden & Lockyer, 1985; Scotto di Perta et al., 2019). Previous studies found that the difference between chamber and micrometeorological measurements can primarily be
attributed to air-side mass transfer and rainfall (Eklund, 1992; Hafner et al., 2023.; Smith & Watts, 1994; Sommer & Misselbrook, 2016). It is likely that it is also the case in this study, as the measurements were conducted in the same field plot after farm machine application, hence the differences cannot be attributed to differences caused by slurry, application, or soil. During both trials with parallel bLS measurements, the ambient wind speed was relatively low (Fig. 4), and during trial C slight precipitation also occurred, which likely contributed to low emissions for the bLS measurements.

Considering uncertainty in measured emission based on random error for DFC and concentration bias for bLS, both the reduction in emission (injection compared to trailing hose) and emission following injection were similar between the two methods. Both methods showed low emission after injection. For DFC measurements, the 95% confidence interval was 2.3 to 7.1% of applied TAN ($n$ = 7 chambers), compared to -1.8 to 2.4% of applied TAN for bLS for injection. For trailing hose application, the intervals were 37% to 49% of applied TAN ($n$ = 6 chambers) for DFC and 9.9% to 15% of applied TAN for
bLS. The NH$_3$ emission reduction obtained by injecting the digestate compared to application by trailing hose was measured to be 89% with DFC measurements, with a 95% confidence interval of 83 to 95% (or 85% to 93% from the bootstrap approach). For bLS, the reduction was 97% relative to trailing hose emission. Uncertainty for bLS was 1.2% of applied TAN ($s_{emis}$), which is a relatively small value, but large compared to a loss of 0.23%, resulting in a confidence interval of 74% to 100% (assuming a reduction > 100%, representing uptake of NH$_3$ after slurry application, which is not plausible). Thus, the range
for reduction obtained by injection is similar for DFC (83%-95%) and bLS (74%-100%). Uncertainty in bLS results here reflect challenges in measuring a small concentration increase compared to background level, which was a consequence of a small plot with very efficient injection. Using a small plot means that the source area was very small compared to the concentration footprint area (Kamp et al., 2021), this affects the number of touchdowns inside the source area for the bLS model, which is enhanced in some wind directions due to the rectangular shape of the plot.

**4 Conclusions**

A new dynamic flux chamber measurement system with online measurements showed lower variability than an earlier generation system of wind tunnels. For trials with trailing hose application, variability increased drastically from manual (handheld) application (coefficient of variation, cv = 5%) to application with 3-m slurry boom (14% cv), and even further for application with a 30-m farm-scale slurry boom (20% cv). This increase in variability can potentially make it difficult to
measure small differences between treatments when slurry is applied by machine, unless larger sample sizes (more repetitions) are used. The differences in variation between handheld application and machine application should be taken into consideration when planning field trials. The flux measured with the DFC was consistently higher than bLS measured flux from the same plot in two experiments, with differences likely caused by the relatively high air exchange rate in the DFC. Due to this

difference the new DFC with the chosen AER should not be used for estimation of open-air emissions but can be used to
estimate relative differences between application methods or slurry treatments.

**cRediT authorship contributions statement:**

J. Pedersen: Conceptualization, Methodology, Validation, Formal analysis, Investigation, Data curation, Writing – Original
Draft, Writing – Review & Editing, Visualization, Supervision, Project administration; S. D. Hafner: Conceptualization,
Methodology, Writing – Review & Editing; V. I. Karlsson: Software, Validation, Formal analysis, Writing – Review &
Editing; L. Rong: Software, Validation, Formal analysis, Writing – Original Draft, Writing – Review & Editing; A. Pacholski:
Conceptualization, Methodology, Writing – Review & Editing; R. Labouriau: Methodology, Formal analysis, Construction
and conception of the Gamma-based generalized linear mixed model, Writing – Review & Editing, Visualization; J. N. Kamp:
Validation, Formal analysis, Investigation, Data curation, Writing – Original Draft, Writing – Review & Editing, Visualization

**Competing interests**

The contact author has declared that none of the authors has any competing interests.

**Acknowledgements:**

The authors would like to thank the technical staff at Aarhus University Heidi Grønbæk, Jens Kr. Jensen, Leonid Mshanetskyi,
and Peter Storgård Nielsen for their skillful construction of the dynamic flux chambers, carrying out the field trials and
laboratory analysis, Jens Bonderup Kjeldsen for drone pictures, and Søren Anton Kirk Nielsen and Arne Grud for digestate
application. The authors are grateful to Søren Mejlstrup Jensen from Samson Agro A/S for providing equipment for application
of digestate in the 30-m plot and useful discussion.

**Funding:**

This work was funded by the Ministry of Food, Agriculture and Fisheries of Denmark through two green development and
demonstration programs (GUDP) with the project titles *Methods for reducing ammonia loss and increased methane yield from*
*biogas slurry (MAG)* (journal nr.: 34009-21-1829) and *eGylle* (journal no.: 34009-20-1751).

**Data availability:**

https://doi.org/10.5281/zenodo.8424778




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
