# Peer review of "Optimized design of flux chambers for measurement of ammonia emission after field application of slurry with full-scale farm machinery"

_Atmospheric Measurement Techniques, 2023_

## Author Response (AR1)

Dear reviewers,

Thank you for your valuable comments.

Please find below our replies and actions on your specific comments.

**RC1**

- Comment: Paragraph 3.1: please could you better clarify the geometry of the deflector plate and its function.
  Reply: The geometry of the deflector plate is described in section '2.1.1 Chamber design'. This section also refers to Fig. 1 and S1 showing sketches of the DFC. The following sentence has been added to paragraph 2.1.1 to clarify the function of the deflector plate: 'The design with a deflector plate was chosen in order to attempt to distribute the inflow air evenly above the emitting surface'.
- Comment: Lines 309-311: please add some references to this section in order to enhance the discussion part related to the CFD.
  Reply: A reference and some discussion has been added.
- Comment: Lines 318-319: When you wrote about the final inlet design, could you describe it?
  Reply: The inlet design is thoroughly described in section 2.1.3 (*The new inlet design consisted of 100 mm PTFE tube (OD: 6.35 mm, ID: 4.75 mm) inserted into a 15 mL plastic centrifuge tube which was itself inserted into a 50 mL plastic centrifuge tube. All three (PTFE tube and both centrifuge tubes) had 3 rows of 5 small holes (Fig. S3)*). A reference to this section has been added to section 3.2. discussing the inlet designs.
- Comment: Figure 4C: Please explain the reason why there are so many differences among DFC replicates in the last trial.
  Reply: Please note that the y-scales in Fig. 4 differ, the following has been added to the figure text: 'Please note that y-scale differs'. For 'trial C, Injection' the sd of the replicates is very low, but as the emission is also very low the cv is large (Table 4), implications of the variation among replicates are discussed in section 3.2.2. The variation is smaller than in trial B. We don't think it is possible to draw further conclusions than the ones already discussed based on the small dataset.
- Comment: When you wrote 'During both trials with parallel bLS measurements, the ambient wind speed was relatively low' is it possible that the air speed inside the chambers did not fit the wind speed of the external environment at the same height? Maybe it is higher, that is why you may have an overestimation of the NH3 emissions?
  Reply: We agree that the higher air speed inside the DFCs most likely caused the higher emissions measured with DFC compared to bLS, this is also stated in the same section: '*Previous studies found that the differences between chamber and micrometeorological measurements can primarily be attributed to air-side mass transfer and rainfall (refs). It is likely that it is also the case in this study, as the measurements….*'.

**RC2**

- Comment 1: Objectives of the work are presented at the end of the introduction but then a long statement on CFD (lines 50-60). Maybe better moving that text above thus clearly showing the objectives of the work for the readers.
  Reply: The last section has been reorganized to make the objectives of the work clear, and a part of the CFD description has been moved down to section 2.1.2.
- Comment 2: In subsection 2.1.1 the sentence 'Inspired by the laboratory chambers used by Dominique et al. (2013) and Georgios et al. (2013) is just a repetition of the first sentence in section 2. Please, try to modify it to avoid being redundant.
  Reply: To avoid unnecessary repetition the first part of the sentence in 2.1.1 has been deleted.
- Comment 3: Please, add the author/source of the figure 1. Same for figure 2.
  Reply: All figures are original creations of the authors. Source information can be provided if needed.
- Comment 4: Line 160: 'Anaerobically digested slurry', cow slurry?
  Reply: The digestate was produced from several sources of slurry and other organic materials. In section 2.2.2 a detailed description is provided: *The digestate was produced at the biogas plant at Aarhus University, which operates two reactors in series at 51°C for 14 d and 47°C for 40 d. After the second reactor, the digestate was pumped to a concrete storage tank, where the digestate for the trials was collected. The input to the first reactor in the period where the digestate was produced for the trials was 82% mixed cattle and pig slurry, 9% deep litter, and 9% grass and grass silage (by fresh mass).*
- Comment 5: All tables. Please check and use the same type of letter than the full text.
  Reply: All the tables have been checked and corrected to have same font and font size as full text.
- Comment 6: Line 234. Please, check if there is a grey area in the reference.
  Reply: The grey area has been removed.
- Comment 7: Line 246. Please, check the size of the letters in the case of the reference.
  Reply: The font size of the reference has been corrected.
- Comment 8: Line 279. The year is missing for the reference used.
  Reply: The year has been added.
- Comment 9: Reference list: be uniform with the style and size of letters for all references.
  Reply: The reference list has been thoroughly checked and the errors have been corrected.

---

## Author Response (AR2)

Dear Daniela Famulari,

Thank you for your valuable comments.

Please find below our replies and actions on your specific comments.

Comment: The manuscript "Optimized design of flux chambers for measurement of ammonia emission after field application of slurry with full-scale farm machinery" reports over a study on a novel setup for enclosure measurements of gaseous NH3 emissions from slurry. I would suggest a change in the title "Evaluation of an optimized design of flux chambers for…". In the sense that the described apparatus has been evaluated against models and in a field trial for different conditions: it is just a suggestion, up to the authors to decide.

Reply: We agree that adding 'Evaluation of' to the title will make the title match the content of the manuscript more closely. The title has been changed to: *Evaluation of optimized flux chamber design for measurement of ammonia emission after field application of slurry with full-scale farm machinery.*

Comment: L15: "To evaluate mitigation options, reliable measurements of effects are needed". I am not sure about this sentence: effects of NH3 on the environment? Effects of different practices on NH3 emissions?

Reply: The sentence has been edited to clarify the meaning.

Comment: Fig.1a resolution results inadequate, make sure in the final version you have high resolution figures.

Reply: High resolution figures will be uploaded with the final version.

Comment: L90: AER acronym is defined only later in text, correct that.

Reply: The error has been corrected.

Comment: L212 what is a stream? An air-line?

Reply: 'stream' has been corrected to 'air stream'

Comment: L214-217 and from here on: AER measured in min-1? I don't understand this.

Reply: An explanation of the AER units has been added the first time AER is mention, around L 90: *… to control and measure the volumetric air exchange rate (AER, $m^3$ flow per minute per $m^3$ chamber volume),…*

Comment: L309-311: This is something also the reviewers pointed out (in a different light): I understand the intention to create well mixed conditions within the enclosure; however, that affects

the rate at which NH3 will volatilise from the surface, won't it? Perhaps to justify the regimes of "wind" within the chamber, it would be useful to mention here what are the real wind conditions on the field, to justify the regime chosen for operation? Just a suggestion.

Reply: We added some details on this matter in section 2.1.2: *It was not a goal to mimic ambient wind velocities or mass transfer, as these vary greatly. The chambers are designed to have a constant AER during an experiment in order to keep the measuring system simple, and therefore more robust for field measurements.*